# Project Proposal: Agent for Oversea Marketing Support

**Zhijie Shen, Qian Wu, Keyu Shen**
{shenzj24, q-wu24, sky24}@mails.tsinghua.edu.cn

## Abstract

Traditional methods for overseas marketing require extensive manual efforts, such as reading documents, conducting interviews, following news, and analyzing statistical reports about target countries. In this project, we propose an agent-based system utilizing Large Language Models (LLMs) to enhance the efficiency and intelligence of country studies.

## 1 Background

With the rapid globalization of Chinese companies, understanding international markets is becoming increasingly crucial. Comprehensive country studies are essential to assess market demand, navigate regulations, develop localization strategies, and manage risks. However, traditional methods have several limitations that hinder timely and cost-effective decision-making.

### 1.1 Traditional oversea marketing problem

The current approach involves the following issues:

- **High Costs**: The Ministry of Commerce (MOFCOM) ,commissions the Chinese Academy of International Trade and Economic Cooperation and embassies to generate country reports [1]annually, consuming significant financial and human resources.
- **Outdated Information**: These reports only include data from the previous year, lacking real-time insights.
- **Limited Usability**: The reports are typically in paper format, making it challenging to search for specific information or conduct in-depth analyses like competitor profiling, product feedback analysis, and potential user identification.

## 2 Related work

Global Market Research Companies like Nielsen, Gartner, and Forrester Research employ AI and LLMs to process vast amounts of data for global market insights. They analyze trends, customer sentiment, and competitive landscapes to provide detailed reports that assist businesses in making informed decisions about overseas markets.

The implementation of this project is already supported by the following technical foundations:

### 2.1 Crawler Tools

- **Web Web Scraping** : Tools like BeautifulSoup or Scrapy in Python are used to extract data from websites.

---

[1]http://fec.mofcom.gov.cn/article/gbdqzn/

Preprint. Under review.

- **Social Media API**:Data can be collected from platforms like Twitter, Facebook, or regional social networks to provide timely insights.
- **Data Providers**: Utilize services that offer datasets for specific countries or industries.

## 2.2  LLM Integration

Open-source LLM models will be employed to analyze data and generate country insights, enabling automated, accurate analysis of complex information.

## 2.3  OSINT experience

Through my professional experience, I have gained substantial expertise in OSINT, particularly in the use of various tools.

# 3  Proposal method

The proposed AI agent will streamline the process of country studies by generating comprehensive guides and supporting overseas marketing tasks like competitor analysis and customer insight generation.

we plane use LangChain as AI agent Frameworks.

1. **Data Ingestion**: The agent will collect data from specified sources, including online databases, social media, and official reports.
2. **Data Cleaning**: Preprocessing techniques will remove noise and irrelevant information, extracting key information for structured storage.
3. **Analysis**: The AI agent will utilize LLMs to analyze structured data, focusing on trends, competitor behavior, and user sentiment.
4. **Output Generation**: The system will produce summaries, detailed reports, and visualizations for user-friendly insights.
5. **Feedback Loop**: Incorporate mechanisms for user feedback to refine the agent's performance.
6. **Regular Task Scheduling**: Tasks will be scheduled to run at regular intervals, ensuring up-to-date informations.
7. **Enhanced OSINT Integration**: The workflow will be improved with more advanced OSINT techniques, making the agent more versatile.

