# OpenReview forum: "【Proposal】 Agent for Oversea Marketing Support"
_tsinghua.edu.cn/THU/2024/Fall/AML — THU 2024 Fall AML Submission_

### Official Review · ~Feihong_Zhang1 · 2024-11-06

**Rating:** 4
**Confidence:** 2

**Review:**

No references are listed, many of them seem to be generated by GPT, the idea is not very clear, and the scheme is not fully understood.

---

### Official Review · ~Yunghwei_Lai1 · 2024-11-08

**Rating:** 4
**Confidence:** 3

**Review:**

No references and insufficient elaboration on methodology. The whole proposal is too vague.

---

### Official Review · ~Renrui_Tian1 · 2024-11-09
**Clear Problem Identification, but Methodology and Writing Require Refinement**

**Rating:** 4
**Confidence:** 4

**Review:**

**Strengths**:
* **Identifies a clear problem**: The proposal effectively highlights the limitations of traditional overseas marketing methods, particularly the reliance on outdated reports and manual data analysis.

**Weaknesses**:
* **Vague and underdeveloped methodology**: The proposed workflow lacks specific details on data collection methods, LLM model selection and training, data cleaning techniques, and analysis methodologies.
* **Lack of specificity in data sources**: The proposal mentions data collection from online databases, social media, and official reports but does not specify which sources will be used. This lack of detail raises concerns about the comprehensiveness and reliability of the data.
* **Lack of clarity on LLM utilization**: The proposal mentions the use of open-source LLM models but lacks details on specific models, potential fine-tuning requirements, and how the models will be integrated into the agent's workflow.
* **Lack of bibliography and references**: The proposal does not include a bibliography or references section, making it difficult to verify the sources of information and research mentioned.
* **Grammar and style issues**: The proposal contains some obvious grammatical errors and inconsistencies, impacting its rigor and professionalism.

**Overall**: The proposal presents a general idea for utilizing AI and LLMs in overseas marketing support. However, it lacks sufficient detail, clarity, and consideration of critical aspects such as data sources and LLM utilization. Addressing these weaknesses is crucial for developing a viable and impactful solution.

---

### Official Review · ~Tim_Bakkenes1 · 2024-11-09

**Rating:** 5
**Confidence:** 3

**Review:**

The problem highlighted by the proposal is very clear and the benefit of utilising agents for oversea marketing support could have a big impact on it. Breaking down the problems with current approach into 3 main parts, as you do, strengthens the potential of your research.

There are some issues with the proposal.

1. The grammar: "we plane use LangChain as AI agent Frameworks" - This sentence should be: "We plan to use LangChain as an AI agent Framework". There are more instances where the grammar is awkward or incorrect. Make sure to proofread before you submit.

2. References: It would improve the quality of the proposal if it was built on more trusted sources.

3. Methodology: The method proposed in the proposal fails to give examples on what data sources can be used to get data from websites and what LLM:s that could be utilised. While the method is broad, it would benefit from a bit more depth.

Overall the proposal presents an interesting research topic but could be refined to make it more clear for the reader how these agents could be developed.

---

### Official Review · ~Guangjie_Xu1 · 2024-11-09

**Rating:** 5
**Confidence:** 3

**Review:**

The proposal plan to address a relevant and impactful issue—enhancing the efficiency of overseas market research for Chinese companies.

**Pros**

1. The planned steps—data ingestion, cleaning, analysis, and output generation—are logical and align well with the project’s goals.

**Cons**

1. While aiming for timely information, the proposal does not thoroughly explain how it will manage real-time data challenges.
2. The proposal could benefit from providing more specific examples or use cases to illustrate the agent’s applications in overseas marketing tasks.
3. No references

---

### Official Review · ~Wenjing_Wu1 · 2024-11-11

**Rating:** 5
**Confidence:** 3

**Review:**

**Summary**:
The proposal outlines a method based on Large Language Models (LLMs) designed to address challenges in traditional overseas market research.

**Strengths**:
- **Clear Problem Identification**: The proposal effectively describes the limitations of traditional overseas market research approaches, providing a clear context and rationale for the new method.
- **Logical Structure in Methodology**: The methodology section is organized in a step-by-step format, which makes it easy to follow and logically sound.

**Weaknesses**:
- **Formatting Issues**: The proposal’s format diverges from the conventional structure. For instance, there is only a "1.1" subsection without a subsequent "1.2", which appears unnecessary and disrupts the document's flow.
- **Issue in Organizational Structure**: The abstract seems to function as the introduction, which could lead to confusion.
- **Lack of References**: The proposal would benefit from a set of relevant references to strengthen its credibility and demonstrate familiarity with existing literature.
- **Insufficient Detail in Methodology**: While the methodology is logically organized, it lacks concrete details for each step. Adding specific information would enhance the clarity and depth of the approach.

---

### Official Review · ~Yang_Ouyang2 · 2024-11-11
**Good structure and understanding, but lack details**

**Rating:** 7
**Confidence:** 3

**Review:**

Strengths:

Relevant Solution: Addresses traditional marketing research limitations such as costs, outdated information, usability
Well-Defined: Authors have a clear understanding in the challenges of traditional overseas marketing methods.
Structured Methodology: Authors provide a clear workflow, including data ingestion, analysis, and feedback.

Weaknesses:

Lacks Specificity in LLM Application: Lacks detail on how LLMs can diverse datasets.
Unclear LangChain usage: How does it help?
Data Privacy: Since the Agent collects data, is the data collected allowed?

---

### Official Review · ~XueZeng1 · 2024-11-11

**Rating:** 4
**Confidence:** 3

**Review:**

This proposal shows the clear steps to solve the problem.

However,there is no in-depth thinking about the technical challenges encountered in solving this problem, such as the ability of LLM to deal with problems in a specific field,and no references.

---

### Official Review · ~Maanping_Shao1 · 2024-11-12

**Rating:** 6
**Confidence:** 3

**Review:**

This proposal outlines a promising approach to streamline overseas marketing through an AI-driven agent utilizing Large Language Models (LLMs). The project aims to address the high costs, outdated information, and limited usability associated with traditional market analysis methods. By incorporating LLMs and tools like web scraping and social media APIs, the proposed agent will enable real-time data ingestion, analysis, and output generation, enhancing decision-making efficiency for international market insights.

The methodology is well-defined, focusing on automated data collection, preprocessing, analysis, and reporting, with a structured feedback loop to improve performance. This approach could significantly reduce costs and provide up-to-date, actionable insights, making it a strong candidate for further development. However, clarity on specific challenges in LLM-based analysis, data privacy, and accuracy metrics would strengthen the proposal.